# Local Food Environment and Consumption of Ultra-Processed Foods: Cross-Sectional Data from the Nutritionists’ Health Study—NutriHS

**DOI:** 10.3390/ijerph20186749

**Published:** 2023-09-13

**Authors:** Brena Barreto Barbosa, Lucca Nielsen, Breno Souza de Aguiar, Marcelo Antunes Failla, Larissa Fortunato Araújo, Larissa Loures Mendes, Soraia Pinheiro Machado, Antonio Augusto Ferreira Carioca

**Affiliations:** 1Postgraduate Program in Nutrition and Health, Ceará State University, Fortaleza 60714-903, CE, Brazil; brena.barreto@aluno.uece.br (B.B.B.);; 2Coordination of Epidemiology and Information, Municipal Health Department of São Paulo, São Paulo 01002-900, SP, Brazilmarcelofailla@prefeitura.sp.gov.br (M.A.F.); 3Graduate Program in Public Health, Department of Community Health, Faculty of Medicine, Federal University of Ceará, Fortaleza 60020-181, CE, Brazil; 4Postgraduate Program in Nutrition and Health, Department of Nutrition, Federal University of Minas Gerais, Belo Horizonte 31270-901, MG, Brazil; 5Nutrition Course, Health Sciences Center, University of Fortaleza, Fortaleza 60811-905, CE, Brazil

**Keywords:** food consumption, food processing, healthy eating, nutritionists

## Abstract

This study aimed to analyze whether community food environments are associated with individual food consumption among nutrition students and newly graduated nutritionists. This cross-sectional study used data from the Nutritionists’ Health Study cohort, which included 357 undergraduate nutrition students from the city of Fortaleza, Ceará, Brazil. Exposure to the food environment was defined as the proximity and availability of food outlets within a 500 m buffer from the participants’ homes. Food consumption was assessed using a Food Frequency Questionnaire and analyzed according to the NOVA classification. Multi-level linear regression models with fixed effects were used to estimate the presence of food outlets within the buffer and their association with food consumption. The presence of mini-markets in the buffer in the fourth quartile was associated with lower consumption of ultra-processed foods (UPF) when observing socioeconomic and lifestyle conditions (β = −3.29; 95% CI = −6.39 to −0.19). The presence of bakeries and coffee shops was related to lower consumption of ultra-processed foods among participants when observing socioeconomic conditions (β = −3.10; 95% CI = −6.18 to −0.02). There was no clear evidence of an association between the type of food outlet and UPF consumption. The community food environment seemed to influence food consumption among study participants, although clearer and more consistent evidence on this subject is needed.

## 1. Introduction

Food processing is a potential influencer of obesity [1] and other health conditions [2,3]. The categorization of foods based on the extent and purpose of processing was proposed based on the NOVA classification. Foods were subdivided into four groups: unprocessed or minimally processed foods, processed culinary ingredients, processed foods, and ultra-processed foods (UPFs) [4]. Ultra-processed foods are ready-to-eat products that contain substances and additives used in the manufacture of processed foods such as sugar, salt, and oil, in addition to stabilizers and preservatives that enhance the flavor, aroma, and texture [4]. UPFs account for a high percentage of food consumption in both middle- and high-income countries [5]. This is attributed to the low cost and increased global availability of these products, which cause significant changes in the environment and dietary patterns [6].

Food environment refers to the interface between consumers and the food system, encompassing the availability, accessibility, convenience, and desirability of food [7]. It can be categorized into four components: consumer food, information food, community food, and organizational food environments [8].

The community food environment, which includes the number of establishments selling food in a region, their location, and residents’ accessibility to the types of services offered [8], has undergone relevant changes in recent decades. The greater supply and accessibility of foods, especially ultra-palatable, ultra-processed foods, has a negative impact on food consumption patterns in most countries [9], which contributes to the community food environment being identified as a favorable or hindering factor for access to these products, depending on their characteristics [10,11,12].

Despite evidence indicating that people with higher education are more likely to have a diverse and healthy diet [13], university students exhibit unhealthy eating behaviors. Even among undergraduates in health sciences such as nursing, medicine, and nutrition, high consumption of fast foods, sweet drinks, soft drinks, and alcoholic beverages is observed, as well as low consumption of fruits, vegetables, fish, whole grains, and legumes [14].

Although studies have provided evidence that a community’s food environment may be associated with nutritional status [15,16,17] and food consumption [18,19], there is little research on the association between the food environment and consumption of processed foods [20]. This investigation is of great importance because it explores the connection between consumption patterns based on food processing and their impact on health and disease conditions [20,21]. Considering the relevance of the food environment and individual food choices, in addition to the scarce evidence available in the literature on the subject, this investigation is expected to contribute to our understanding of the association between the local food environment and food consumption in adults. To accomplish this, a cohort of nutrition students and newly graduated nutritionists was used as the study population, which is an advantage of this investigation. These individuals are health professionals who engage in dietary assessments on a regular basis, enabling them to provide more pertinent and reliable food consumption data. Thus, this study aimed to analyze whether the community food environment influences food consumption among nutrition students and newly graduated nutritionists.

## 2. Materials and Methods

This is a cross-sectional study using data from the Nutritionists’ Health Study—NutriHS (Nutritionist Health Study) cohort [22], carried out between 2018 and 2019 in the city of Fortaleza, Ceará, Brazil. NutriHS is a multicenter cohort study coordinated by universities around Brazil: the University of São Paulo in the city of São Paulo; the State University of Campinas in the city of Limeira; and the University of Fortaleza and Ceará State University in Fortaleza. It was carried out with undergraduate nutrition students and trained nutritionists, with the main objective of evaluating the relationship between diet, intestinal microbiota, and cardiovascular risk.

All those who agreed to participate were informed about the research objectives and expressed prior agreement through the free and informed consent form. This study was approved by the Ethics Committee of Ceará State University under opinion number 95402618.3.0000.5534.

### 2.1. Study Scenario and Sample

Fortaleza, with an estimated population of 2.6 million inhabitants, is the fifth most populous city in Brazil. It has a total area of 312,353 km^2^, with a Municipal Human Development Index (MHDI) of 0.754 [23] (Figure 1). MHDI is a composite measure of indicators of three dimensions of human development: longevity, education, and income. The index ranges from 0 to 1. The closer it is to 1, the greater the human development [23]. During this study, the municipality included 119 districts and was divided into six administrative regions managed by the Regional Executive Secretariats (SER). These regions consist of districts with geographic proximity and similar socioeconomic characteristics.

This study was carried out with nutrition students and recently graduated nutritionists. This approach was chosen to minimize follow-up losses in the NutriHS cohort study and to leverage a unique opportunity for gathering high-quality dietary information. The minimum sample size for NutriHS was previously estimated, considering the detection of a correlation coefficient ≥ 0.20 (alpha, 5%; beta, 20%), with an additional 20% increase to account for follow-up losses. Therefore, a minimum of 233 volunteers was required for each center.

Fortaleza has nine Higher Education Institutions (HEIs) that offer undergraduate courses in nutrition [24]. This research was coordinated by two HEIs located in the city of Fortaleza, namely, the University of Fortaleza and Ceará State University. Undergraduate nutrition students from seven HEIs in the city participated in the study and signed the letter of consent from the Research Ethics Committee. The recruitment of participants for the study was conducted through project dissemination on social networks (Facebook and Instagram) and in-person communication at the HEIs. Interested students were directed to the project’s online page (http://www.fsp.usp.br/nutrihs, accessed on 1 June 2021) to fill in the questionnaires.

A total of 734 students registered on the site. However, 153 participants did not complete the Food Frequency Questionnaire (FFQ) and were excluded from the analysis. Additionally, outliers with caloric intake below 500 or above 5000 kcals were excluded in order to reflect more realistic consumption patterns, following the approach proposed by Willett [25]. Subsequently, 224 participants who either did not reside at a permanent address in the city of Fortaleza or registered with incomplete addresses were also excluded. As a result, a total of 357 participants remained eligible for inclusion in the study (Figure 2). There were no sociodemographic differences between the 734 participants who registered on the website and the 357 participants in the final sample, with data losses referring to randomly missing data.

### 2.2. Data Collection and Processing

Information on sociodemographic and health characteristics, physical activity, and food consumption was collected and used in the present study.

#### 2.2.1. Sociodemographic Characteristics

The variables collected included the following: gender (male/female), date of birth for age calculation in years, full address (number, street/avenue name, complement, city, state, and postal address code), university affiliation and city of the university, skin color (white, black, brown, East Asian, and indigenous), education of the head of the family (never attended school/incomplete 1st grade, complete 1st grade, complete 2nd grade, and university), and family income (<1 minimum wage, 1–5 minimum wages, 6–10 minimum wages, and >10 minimum wages). The minimum wage for the year 2018 was considered, which was BRL 954.0 [26], equivalent to USD 247 during that period.

#### 2.2.2. Anthropometric Data

Participants self-reported their body weight and height, with weight recorded in kilograms and height recorded in meters. Body mass index (BMI) was calculated using the standard formula: weight in kilograms (kg) divided by height in meters squared (m^2^). The BMI values were used to classify weight status—not overweight/obese (BMI < 25 kg/m^2^) and overweight/obese (BMI ≥ 25 kg/m^2^)—following the guidelines of the World Health Organization [27]. Previous studies have shown that self-reported weight and height are reliable for calculating BMI in adults [28,29].

#### 2.2.3. Physical Activity

To estimate physical activity levels, the short-validated version of the International Physical Activity Questionnaire (IPAQ) was utilized. The short version of the IPAQ consists of seven open-ended questions and provides information enabling the estimation of time spent per week engaging in various dimensions of physical activity, including walking, moderate-intensity physical exertion, vigorous-intensity physical exertion, and time spent in a sedentary position (sitting) [30]. The data obtained from the questionnaire were interpreted based on the guidelines provided by the World Health Organization (WHO), which recommends a minimum of 150 min of moderate to intense physical activity per week for adults. Physical activity levels below 150 min per week are considered inadequate according to these guidelines [31].

#### 2.2.4. Food Consumption

The assessment of eating habits was conducted using a previously validated quantitative food frequency questionnaire (FFQ) to estimate the habitual food consumption of adults [32]. This FFQ includes the eating habits of individuals over a year, identifying the usual individual portion of 136 food items, with frequency options (0 to 10 times), time units (day, week, month, or year), and portions (described in household measures, grams and milliliters). Data were processed by the Nutrition Data System for Research (Nutrition Coordinating Center, University of Minnesota, Minneapolis).

Food consumption was evaluated using the NOVA classification (in percentage (%) of total energy), which categorizes foods into four groups according to their level and purpose of processing. These groups include fresh and minimally processed foods, culinary ingredients, processed foods, and ultra-processed foods. A previous publication provides detailed information regarding the categorization within the NOVA classification [4].

Based on the FFQ, the foods were classified into 74 unprocessed or minimally processed foods, 3 culinary ingredients, 19 processed foods, and 40 ultra-processed foods. Using this classification, the percentages of each group’s contribution to daily energy intake were calculated, following the methodology employed in previous studies [33,34].

#### 2.2.5. Local Food Environment

The database of food retail establishments was built using information from two government sources: the Health Surveillance Sector (VISA) of the Municipal Health Department of Fortaleza and the SER of Fortaleza City Hall. VISA, through the Municipal Health Department, conducts regular supervision of the food market through investigative inspections, documentation maintenance, and control records; therefore, it is considered a reliable source of official data [35]. From these databases, information was extracted on the type of establishment and complete address (including number, street name, neighborhood, and postal code) of all licensed food retail stores operating in 2018 and 2019.

The establishments in the database were categorized by VISA according to their main activity, using the National Classification of Economic Activities (CNAE) as a reference. The categories include the following: retail trade of goods, with a focus on food products (hypermarkets, supermarkets, mini-markets, grocery stores, and warehouses); retail trade of specific products (dairy products, cold cuts, sweets, candies, chocolates, meat, fresh produce, and convenience store merchandise); wholesale trade of various food products (milk and dairy products, processed cereals and legumes, fruits, vegetables, roots and tubers, beef, pork and derivatives, fish and seafood, meat and other animal derivatives, breads, cakes, cookies, pasta, ice cream, and confectionery chocolates); restaurants and similar establishments; snack bars, tea houses, juice shops, and similar establishments; bakeries, pastry shops, and ice cream parlors; and coffee shops and street food services.

As a supplementary measure, conferences were conducted to verify the addresses of snack bars (since they are more numerous and subject to greater variability in opening and closing times) using the Google Street View tool. Establishments that presented inconsistencies in the databases and could not be identified through Google Street View were excluded from the study. Information on the MHDI, average per-capita income, and number of residents in the neighborhoods of the city of Fortaleza were extracted from the 2010 Demographic Census [36].

In order to simplify data analysis, all identified food stores were categorized based on an adapted grouping proposed by Costa et al. [37], which takes into account the physical structure of the place, the primary products sold, and specific commercialization characteristics.

In Brazil, there is no consensus on the type of categorization of establishments used to assess the food environment. The categorization method employed in this article, which considers the physical structure of the establishment, the type of products sold, and marketing characteristics, has been previously utilized in studies conducted in Brazil [16,17,21], albeit with some adaptations. As not all establishments in this study fit into the first classification, a combination of two references was utilized. First, an adapted version from Costa et al. [37] was employed, resulting in six main groups: (1) supermarkets and hypermarkets, (2) small markets (minimarkets, grocery stores, and warehouses), (3) street markets, (4) restaurants, (5) snack bars (snack bars, tea houses, juice shops, pastry, and ice cream parlors), and (6) bakeries and coffee shops. Further, for retail and wholesale establishments that did not fit into these categories, the NOVA food classification was used as it categorizes foods based on the extent and purpose of their processing [4], which is the focus of our study. The remaining establishments were grouped according to the predominance of acquisition and food processing characteristics, according to the classification proposed by the Interministerial Chamber for Food and Nutrition Security (CAISAN) [38], resulting in the following two groups: (7) retail of unprocessed and minimally processed foods (vegetables; cereals and pulses; fruits, vegetables, roots, tubers, vegetables, and fresh vegetables; beef and pork; slaughtered poultry and derivatives; fish and seafood; meat and other animal derivatives) and (8) retail of processed and ultra-processed foods (dairy products and cold cuts; confectionery; breads, cakes, cookies, sweets, candies, bonbons, and others; ice cream; pasta; and food products in general).

#### 2.2.6. Statistical Analysis

Sociodemographic and health variables (gender, age, skin color, education of the head of the family, family income, and BMI), and lifestyle (physical activity level), were expressed in absolute (n) and relative (%) values. The caloric contribution of the NOVA classification was expressed as mean and standard deviation. The number of stores/establishments in a 500 m buffer was expressed as median and interquartile range.

Multilevel linear regression models with fixed effects were used to estimate the regression coefficient (beta) and 95% confidence intervals (95% CI). The caloric contribution of the NOVA classification was considered as a dependent variable (continuous) and the presence of food establishments within a 500 m buffer from the volunteer’s residence as an independent variable. The number of food establishments was distributed as quartile (small markets, restaurants, snack bars, retail of unprocessed and minimally processed foods, and retail of processed and ultra-processed foods), tertile (supermarkets and hypermarkets), or presence/absence of establishments (street markets).

The modeling was carried out in two levels and the administrative regions of the municipality were used as a contextual variable. For model 1, the covariates were sex (nominal), age (continuous), skin color (nominal), education of the head of the family (ordinal), and family income (ordinal); for model 2, the covariates were presence of overweight (dichotomous) and physical activity level (dichotomous). The selection of covariates was based on the literature on univariate analysis. The Hausman test was used to choose between random effects and fixed effects models.

The results were reported as regression coefficients (beta) and 95% confidence intervals (95% CI). All analyses were conducted using Stata software version 13.0 (https://www.stata.com, accessed on 23 August 2023) with a significance level of 5%.

## 3. Results

Among the 357 study participants, there was a predominance of females (79.3%), aged under 25 years (72.3%), self-declared black/brown skin color (57.7%), family income from 1 to 5 minimum wages (61.6%), and education of the head of the family with a high school diploma (43.7%). Most participants had a BMI classification as not being overweight (62.5%) and a level of moderate and intense physical activity of 150 min per week (65.0%). Most lived in region 2 of the city (24.4%) and had a higher caloric contribution of unprocessed and minimally processed foods in their diets (59.3%) (Table 1).

The spatial distribution of food establishments within the 500 m buffer around the participants’ homes is shown in Figure 3.

Regarding the association between the presence of establishments in the buffer and the energy contribution of the consumption of unprocessed foods in the diet, for bakeries and coffee shops, a higher consumption of unprocessed food was observed among participants in the third quartile of distribution of the number of establishments when adjusted for covariates of model 1 (β = 4.82; 95% CI = 0.52 to 9.12) and of model 2 (β = 4.84; 95% CI = 0.55 to 9.12) in relation to the first quartile (Table 2).

In the analysis of the association between the presence of establishments in the buffer with the energy contribution of the consumption of processed foods in the diet, significant values were observed for the presence of snack bars among participants in the third quartile (β = 4.46; 95% CI = 0.65 to 8.27) and fourth quartile (β = 4.53; 95% CI = 0.54 to 8.51) of the distribution of establishments, when adjusted for model 1, in relation to the first quartile, and for participants in the third quartile (β = 4.48; 95% CI = 0.65 to 8.32) and fourth quartile (β = 4.43; 95% CI = 0.44 to 8.42), when adjusted for model 2, in relation to the first quartile. Significant values were also found for supermarkets and hypermarkets among participants in the second tertile, in relation to the first tertile, when adjusted for model 1 (β = 3.34; 95% CI = 0.12 to 6.56) and model 2 (β = 3.26; 95% CI = 0.02 to 6.51); bakeries and coffee shops, among participants in the fourth quartile, when adjusted for model 1 (β = 5.00; 95% CI = 1.25 to 8.74) and model 2 (β = 4.88; 95% CI = 1.13 to 8.64), in relation to the first quartile; retailing of processed and ultra-processed foods among participants in the third quartile (β = 6.54; 95% CI = 2.30 to 10.79) and fourth quartile (β = 4.63; 95% CI = 0.59 to 8.67), when adjusted for model 1, in relation to the first quartile, as well as among participants in the third quartile (β = 6.34; 95% CI = 2.04 to 2.63) and fourth quartile (β = 4.62; 95% CI = 0.54 to 8.70) when adjusted for model 2, in relation to the first quartile; and retail of unprocessed and minimally processed foods among participants in the second quartile, when adjusted for model 1 (β = 4.51; 95% CI = 0.73 to 8.29) and model 2 (β = 4.74; 95% CI = 0.95 to 8.54), in relation to the first quartile (Table 3).

For the association between the presence of establishments in the buffer and the energy contribution of the consumption of ultra-processed foods in the diet, the presence of mini-markets was associated with lower consumption among participants in the fourth quartile of distribution of establishments, when adjusted for model 1 (β = −3.55; 95% CI = −6.65 to −0.45) and for model 2 (β = −3.29; 95% CI = −6.39 to −0.19), in relation to the first quartile. The presence of bakeries and coffee shops was related to lower consumption of ultra-processed foods among participants in the third quartile, when adjusted for model 1, compared with the first quartile (β = −3.10; 95% CI = −6.18 to −0.02) (Table 4).

## 4. Discussion

We identified a statistical association between the presence of food establishments in the buffer around nutrition students’ homes and food consumption in the three food processing groups analyzed. We found an inverse association between the caloric contribution of UPFs and the presence of bakeries and coffee shops, and of small markets close to the participants’ homes. The presence of snack bars, supermarkets and hypermarkets, bakeries and coffee shops, and retail stores of processed and ultra-processed foods close to the participants’ homes showed a significant association with the energy contribution of processed foods. However, despite the statistical association found, there is no clear or strong evidence of an association between the types of food outlets investigated and UPF consumption.

According to the NOVA classification, processed foods are foods manufactured with the addition of salt or sugar, and occasionally oil, vinegar, or another substance, from processed culinary ingredients to food from the unprocessed or minimally processed food group. These foods can be produced by non-alcoholic fermentation, such as cheese and bread, and alcoholic fermentation, as is the case for beverages such as wine, beer, and cider [4]. Processed foods are often purchased in bakeries and confectioneries, hypermarkets, dairy and cold-cut retailers, food retailers in general, and restaurants, as they are considered mixed establishments. Therefore, they are important places to purchase healthy (unprocessed and minimally processed foods) and unhealthy foods (processed and ultra-processed foods) [38].

The purchase of alcoholic beverages may have influenced the association between the presence of establishments in the buffer zone and the consumption of processed foods. Alcohol consumption has become increasingly common among young people [39]. In Brazil, 28.1% of adolescents aged 13 to 17 consume alcohol [40], and early access is associated with chronic patterns of consumption in adulthood [41]. A study of young Brazilians found that 17.54% of respondents reported purchasing alcoholic beverages in establishments such as markets, liquor stores, bars, or supermarkets [39]. Another study found that approximately 10% of adolescents interviewed reported buying alcoholic beverages for consumption in coffee shops, bars, and restaurants close to their homes [42].

The presence of establishments in the group of bakeries and coffee shops close to the participants’ homes was statistically associated with the consumption of unprocessed food. Traditionally, bakeries are establishments with greater demand and availability of processed and ultra-processed foods, such as breads, cakes, cheeses, and sausages [43,44]. However, changes in the food environment have allowed bakeries and confectioneries to add new products to their shelves, such as disposables products, beverages, and fresh foods such as fruits, contributing to the fact that non-baked products represented 38.35% of bakery sales in Brazil in 2020 [43].

In this study, the presence of various establishments, such as mini-markets, grocery stores, warehouses, bakeries, and coffee shops showed a negative statistical association with the energy contribution of UPFs. According to the classification proposed by CAISAN, bakeries are establishments with a preponderance of processed food purchases throughout the national territory. However, small markets can be classified as establishments where the main purchase is of fresh foods or establishments where there is no majority purchase of either fresh or ultra-processed foods, characterizing a more “mixed” purchase (depending on the region or Brazilian state) [38]. In a study conducted in Belo Horizonte, which examined the availability and advertising of food outlets, it was identified that convenience stores, grocery stores, and bakeries had the highest availability and advertising of ultra-processed foods [44].

The presence of bakeries and coffee shops near the participants’ homes was inversely associated with the consumption of UPFs in the third quartile. However, this association was not significant after adjusting for BMI and physical activity, indicating that these factors may mediate the relationship. Furthermore, no association was observed in the fourth quartile (highest) compared with the first quartile, which would be expected if there was a genuine correlation between the density of bakeries in a local area and UPFs consumption. The same was found for the presence of supermarkets and hypermarkets and their association with processed food consumption, which was significant in the second tertile but not in the third tertile. These findings suggest that the relationship between the densities of food establishments and consumption of foods based on the level of processing is non-linear, highlighting the need for further investigation and understanding of the underlying mechanisms. In this study, there was no clear evidence of an association between any type of food establishment and the consumption of ultra-processed foods. In a study that aimed to describe the consumption patterns of UPFs in the Netherlands and to evaluate whether exposure to the food environment was associated with the consumption of UPFs, it was found that greater exposure to restaurants and supermarkets was associated with slightly lower consumption of UPFs [21].

The inverse association found between the caloric contribution of UPFs and the presence of bakeries, coffee shops, and small markets close to their residence may be attributed to the fact that the food choices and consumption of nutrition students and nutritionists can be influenced by knowledge, since they are professionals who, among other attributes, use nutritional knowledge as a way of promoting health and preventing diseases [45]. The main factors that determine food choice are identified as intrapersonal (knowledge, beliefs, and attitudes), biological (innate abilities of individuals), sensory-affective (feelings and emotions in relation to food), and environmental (availability and accessibility of food, which are the easiest to influence) [46]. In a study that aimed to investigate the main determinants of food consumption in a university environment, the price, quality of meal, and location/distance from the place of sale were identified as decisive conditions, with differences between the sexes and the availability of healthy food choices being more important for women than for men [47].

This study has some important limitations that may interfere with the interpretation of the results, such as the fact that the participants self-reported their body weights, which is likely to be biased. In addition, the generalizability of the results of this study is limited for the non-nutrition-related groups, even to non-nutrition-related groups, given the relatively small sample and specificities of the sample of nutrition students. Another relevant issue is that the assessment of food consumption was based on the NOVA classification using an FFQ. Most FFQs are not specifically designed to identify food processing details; therefore, misclassifications are possible [48]. For example, items such as fruit juices, milkshakes, meatballs, hamburgers, or pizza can be consumed as artisanal or industrial varieties. However, only industrial varieties are classified as ultra-processed. Similarly, for yogurt and whole grains, FFQs generally do not distinguish between plain, sweetened, or flavored varieties [49]. Consequently, while certain products may fall under the ultra-processed category, others can be classified as minimally processed [9,50].

The absence of data on food stores in the three districts of the city limits the characterization of the local food environment. Although these areas have food sale establishments, the absence of information on their locations with VISA indicates that many operate informally. The existence of informal food vendors must be acknowledged because they may conceal the true distribution of food establishments in the city. The absence of these data may be due to the absence of the state from more vulnerable spaces and its failure to reach this population.

Using VISA information to build the database may have resulted in an underestimation of the actual number of establishments in the city, given that establishments not registered in any secondary database and street vendors were not included. Addressing these limitations could be achieved by incorporating on-the-spot observations [51]. Nonetheless, by combining the two databases and virtually merging establishments within the snack bar group, more reliable information was obtained for assessing the city’s food environment. However, the lack of data on consumer’s food environments, such as the quality, variety, price, and advertising of available food within the identified establishments, limits a more thorough characterization of the local food environment.

Despite these limitations, the strong point of this study is the analysis of the relationship between food consumption based on food processing and the community food environment. The fact that multilevel linear regression models were used, considering sociodemographic and lifestyle variables, and that they were applied to different types of food establishments, is also a strong point. In addition, the use of a comprehensive FFQ to obtain information on food consumption from nutrition students and nutritionists may have contributed to minimizing the limitations of using this survey as a way of evaluating food processing consumption. This is due to the fact that these individuals are health professionals who are either in training or have already graduated, and they regularly engage in dietary assessments. Consequently, their familiarity and expertise in this area likely enabled them to provide more precise and accurate consumption data.

## 5. Conclusions

The presence of mini-markets and bakeries close to the participants’ homes was statistically associated with a lower consumption of ultra-processed foods. The presence of bakeries and coffee shops is associated with the consumption of fresh food. The presence of supermarkets and hypermarkets, snack bars, bakeries, and processed and ultra-processed retail stores was statistically associated with a higher consumption of processed foods. However, despite this statistical association, there was no clear evidence of an association between the types of food outlets and food consumption. Thus, the community food environment seems to have the potential to influence the food consumption of the study participants, although clearer and more consistent evidence on this subject is needed.

## Figures and Tables

**Figure 1 ijerph-20-06749-f001:**
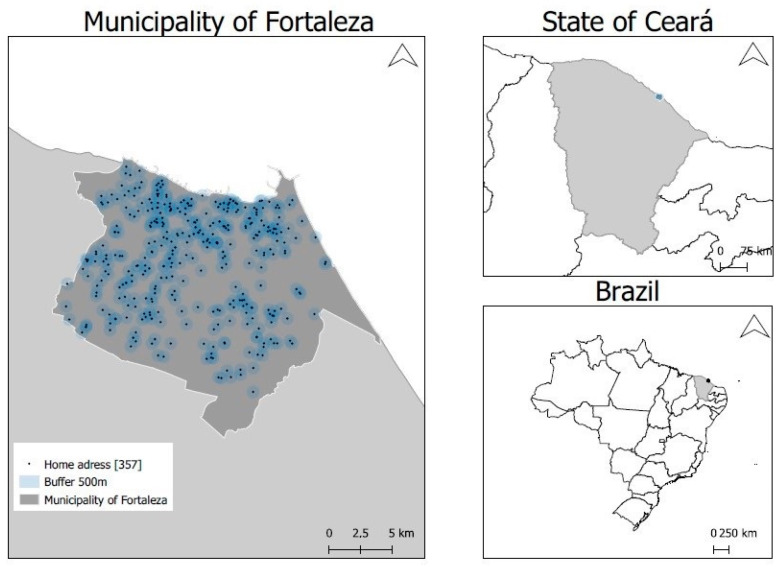
Location of the city of Fortaleza and spatial distribution of the Nutritionists’ Health Study volunteers with a 500 m buffer, Ceará, Brazil.

**Figure 2 ijerph-20-06749-f002:**
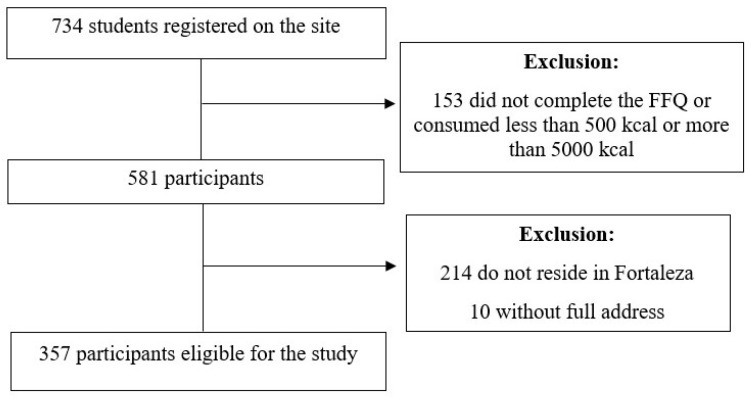
Flowchart for the selection of participants included in the present study, Fortaleza, Brazil.

**Figure 3 ijerph-20-06749-f003:**
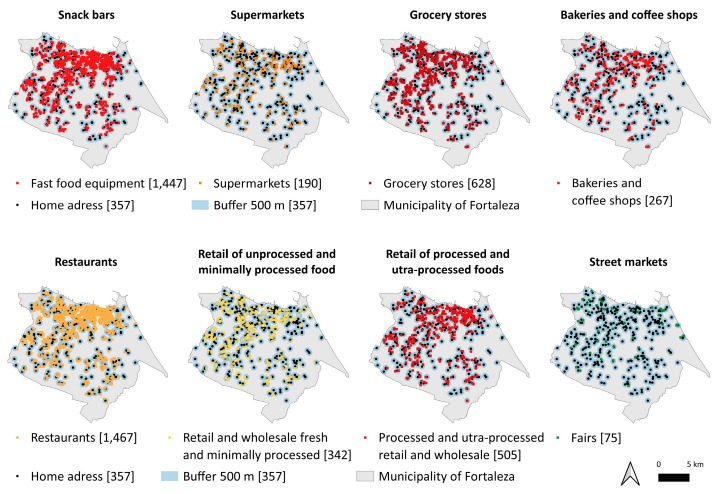
Spatial distribution of food establishments in the 500 m buffer around the participants’ homes. Fortaleza, Ceará, Brazil.

**Table 1 ijerph-20-06749-t001:** Characteristics of the study population, food consumption, and food environment. Nutritionists’ Health Study—NutriHS, Fortaleza, CE.

Variables	N	%
Sex		
Female	283	79.3
Male	74	20.7
Age group		
<25 years	258	72.3
≥25 years	99	27.7
Self-declared skin color		
White	138	38.7
Black/brown	206	57.7
Others	13	3.6
Education of the head of the family		
Never attended school / incomplete 1st grade (from 1 to 7 years of study)	61	17.1
Completed 1st degree (from 8 to 10 years of study)	36	10.1
Completed high school (from 11 to 13 years of study)	156	43.7
University (14 years or more of study)/Postgraduate	104	29.1
Family income		
<1 minimum wage	23	6.4
1–5 minimum wages	220	61.6
6–10 minimum wages	58	16.2
>10 minimum wages	39	10.9
BMI classification		
Non-overweight/obesity	223	62.5
Overweight/obesity	134	37.5
Moderate and intense physical activity level		
<150 min/week	121	33.9
≥150 min/week	232	65.0
Regional		
1	42	11.8
2	87	24.4
3	46	12.9
4	49	13.7
5	60	16.8
6	73	20.5
NOVA classification	Mean	Standard deviation
Unprocessed/minimally processed, % total energy	59.3	14.7
Culinary ingredients, % total energy	3.6	3.6
Processed, % total energy	20.4	12.6
Ultra-processed, % total energy	16.7	10.4
Establishments in a 500 m buffer	Median	p25–p75
Snack bars	6	2–13
Supermarkets and hypermarkets	1	0–2
Small markets	4	1–6
Bakeries and coffee shops	1	0–2
Restaurants	6	2–11
Retail of processed and ultra-processed foods	2	1–5
Retail of unprocessed and minimally processed foods	2	0–3
	N	%
Street markets, ≥1	75	21.0

Values expressed in n and %, mean and standard deviation, or median and p25–p75.

**Table 2 ijerph-20-06749-t002:** Association between the presence of establishments in a 500 m buffer in the energy contribution of unprocessed and minimally processed food consumption in the diet. Nutritionists’ Health Study—NutriHS, Fortaleza, CE.

Energy Contribution from the Consumption of Unprocessed and Minimally Processed Food ^a^				
	Model 1 ^b^		Model 2 ^c^	
	Beta	95% CI	Beta	95% CI
Snack bars (n)				
1° quartile (0–2)	Reference		Reference	
2° quartile (3–5)	−0.84	−5.41, 3.73	−1.14	−5.69, 3.40
3° quartile (6–13)	−2.46	−6.91, 1.99	−2.70	−7.11, 1.71
4° quartile (>13)	−1.56	−6.20, 3.09	−1.27	−5.86, 3.32
Supermarkets and hypermarkets (n)				
1° tercile (0)	Reference		Reference	
2° tercile (1)	−2.09	−5.84, 1.66	−2.16	−5.88, 1.56
3° tercile (>1)	−1.27	−5.08, 2.54	−1.22	−5.03, 2.59
Small markets (n)				
1° quartile (0–1)	Reference		Reference	
2° quartile (2–3)	−1.10	−5.53, 3.32	−0.86	−5.24, 3.51
3° quartile (4–5)	−0.51	−4.90, 3.89	−0.51	−4.87, 3.86
4° quartile (>5)	1.46	−2.93, 5.84	0.88	−3.48, 5.23
Bakeries and coffee shops (n)				
1° quartile (0)	Reference		Reference	
2° quartile (1)	−1.45	−5.77, 2.88	−1.78	−6.09, 2.52
3° quartile (2)	**4.82**	**0.52, 9.12**	**4.84**	**0.55, 9.12**
4° quartile (>2)	−1.84	−6.17, 2.49	−1.54	−5.83, 2.75
Restaurants				
1° quartile (0–2)	Reference		Reference	
2° quartile (3–6)	2.30	−1.96, 6.56	2.30	−1.91, 6.52
3° quartile (7–11)	−1.30	−5.87, 3.27	−1.63	−6.16, 2.91
4° quartile (>11)	−0.05	−4.61, 4.52	0.09	−4.43, 4.61
Retail of processed and ultra-processed foods				
1° quartile (0–1)	Reference		Reference	
2° quartile (1–2)	−2.51	−6.74, 1.72	−2.38	−6.61, 1.86
3° quartile (3–4)	−4.77	−9.73, 0.19	−4.58	−9.53, 0.37
4° quartile (>4)	−3.84	−8.56, 0.88	−3.74	−8.44, 0.96
Retail of unprocessed and minimally processed foods				
1° quartile (0)	Reference		Reference	
2° quartile (1)	−1.55	−5.95, 2.85	−1.75	−6.11, 2.62
3° quartile (2–3)	−2.83	−7.05, 1.38	−3.05	−7.23, 1.13
4° quartile (>3)	−2.55	−6.92, 1.82	−2.51	−6.83, 1.81
Street markets				
No	Reference		Reference	
Yes	−2.83	−6.68, 1.01	−2.83	−6.63, 0.98

Values expressed in beta and 95% confidence interval (95% CI). Values in bold represent statistical significance. ^a^ Multilevel linear regression models with fixed effects. ^b^ Model adjusted for sex, age, skin color, education of the head of the household, and family income. ^c^ Model adjusted for sex, age, skin color, education of the head of the household, family income, body mass index, and physical activity level.

**Table 3 ijerph-20-06749-t003:** Association between the distributions of establishments in a 500 m buffer in the energy contribution of the consumption of processed foods. Nutritionists’ Health Study—NutriHS, Fortaleza, CE.

Energy Contribution from Processed Foods ^a^				
	Model 1 ^b^		Model 2 ^c^	
	Beta	95% CI	Beta	95% CI
Snack bars (n)				
1° quartile (0–1)	Reference		Reference	
2° quartile (3–5)	2.26	−1.66, 6.17	2.41	−1.54, 6.36
3° quartile (6–13)	**4.46**	**0.65**, **8.27**	**4.48**	**0.65**, **8.32**
4° quartile (>13)	**4.53**	**0.54**, **8.51**	**4.43**	**0.44**, **8.42**
Supermarkets and hypermarkets (n)				
1° tercile (0)	Reference		Reference	
2° tercile (1)	**3.34**	**0.12**, **6.56**	**3.26**	**0.02**, **6.51**
3° tercile (>1)	0.78	−2.50, 4.06	0.58	−2.75, 3.90
Small markets (n)				
1° quartile (0–1)	Reference		Reference	
2° quartile (2–3)	2.12	−1.70, 5.94	2.00	−1.82, 5.82
3° quartile (4–5)	2.73	−1.06, 6.53	2.60	−1.21, 6.41
4° quartile (>5)	2.06	−1.73, 5.84	2.39	−1.42, 6.19
Bakeries and coffee shops (n)				
1° quartile (0)	Reference		Reference	
2° quartile (1)	1.34	−2.40, 5.07	1.33	−2.43, 5.10
3° quartile (2)	−1.26	−4.97, 2.45	−1.39	−5.14, 2.36
4° quartile (>2)	**5.00**	**1.25**, **8.74**	**4.88**	**1.13**, **8.64**
Restaurants (n)				
1° quartile (0–2)	Reference		Reference	
2° quartile (3–6)	−0.68	−4.35, 2.98	−0.59	−4.27, 3.09
3° quartile (7–11)	2.50	−1.44, 6.43	2.62	−1.34, 6.58
4° quartile (>11)	3.02	−0.91, 6.95	3.04	−0.90, 6.99
Retail of processed and ultra-processed foods (n)				
1° quartile (0–1)	Reference		Reference	
2° quartile (1–2)	2.45	−1.17, 6.08	2.24	−1.43, 5.90
3° quartile (3–4)	**6.54**	**2.30**, **10.79**	**6.34**	**2.04**, **10.63**
4° quartile (>4)	**4.63**	**0.59**, **8.67**	**4.62**	**0.54**, **8.70**
Retail of unprocessed and minimally processed foods (n)				
1° quartile (0)	Reference		Reference	
2° quartile (1)	**4.51**	**0.73**, **8.29**	**4.74**	**0.95**, **8.54**
3° quartile (2–3)	1.62	−2.00, 5.24	1.63	−2.01, 5.26
4° quartile (>3)	−0.51	−1.56, 0.55	−0.54	−1.61, 0.52
Street markets				
No	Reference		Reference	
Yes	2.66	−0.66, 5.98	2.79	−0.54, 6.12

Values expressed in beta and 95% confidence interval (95% CI). Values in bold represent statistical significance. ^a^ Multilevel linear regression models with fixed effects. ^b^ Model adjusted for sex, age, skin color, education of the head of the household, and family income. ^c^ Model adjusted for sex, age, skin color, education of the head of the household, family income, body mass index, and physical activity level.

**Table 4 ijerph-20-06749-t004:** Association between the distribution of establishments in a 500 m buffer in the energy contribution of the consumption of ultra-processed foods in the diet. Nutritionists’ Health Study—NutriHS, Fortaleza, CE.

Energy Contribution from the Consumption of Ultra-Processed Products ^a^				
	Model 1 ^b^		Model 2 ^c^	
	Beta	95% CI	Beta	95% CI
Snack bars (n)				
1° quartile (0–2)	Reference		Reference	
2° quartile (3–5)	−1.00	−4.24, 2.23	−0.81	−4.06, 2.44
3° quartile (6–13)	−2.84	−5.99, 0.31	−2.63	−5.78, 0.52
4° quartile (>13)	−2.36	−5.65, 0.93	−2.52	−5.80, 0.76
Supermarkets and hypermarkets (n)				
1° tercile (0)	Reference		Reference	
2° tercile (1)	−1.36	−4.02, 1.31	−1.18	−3.84, 1.49
3° tercile (>1)	0.43	−2.28, 3.13	0.56	−2.17, 3.29
Small markets (n)				
1° quartile (0–1)	Reference		Reference	
2° quartile (2–3)	−0.62	−3.74, 2.51	−0.72	−3.83, 2.40
3° quartile (4–5)	−2.64	−5.74, 0.46	−2.53	−5.64, 0.57
4° quartile (>5)	**−3.55**	**−6.65**, **−0.45**	**−3.29**	**−6.39**, **−0.19**
Bakeries and coffee shops (n)				
1° quartile (0)	Reference		Reference	
2° quartile (1)	−0.30	−3.40, 2.80	0.09	−3.02, 3.19
3° quartile (2)	**−3.10**	**−6.18**, **−0.02**	−3.00	−6.08, 0.10
4° quartile (>2)	−2.56	−5.67, 0.54	−2.73	−5.83, 0.36
Restaurants (n)				
1° quartile (0–2)	Reference		Reference	
2° quartile (3–6)	−2.01	−5.03, 1.01	−2.11	−5.13, 0.91
3° quartile (7–11)	−1.49	−4.74, 1.76	−1.32	−4.57, 1.93
4° quartile (>11)	−2.70	−5.94, 0.55	−2.85	−6.09, 0.38
Retail of processed and ultra-processed foods (n)				
1° quartile (0–1)	Reference		Reference	
2° quartile (1–2)	−0.40	−3.42, 2.62	−0.33	−3.37, 2.71
3° quartile (3–4)	−1.75	−5.29, 1.79	−1.67	−5.23, 1.90
4° quartile (>4)	−0.57	−3.94, 2.80	−0.66	−4.04, 2.73
Retail of unprocessed and minimally processed foods				
1° quartile (0)	Reference		Reference	
2° quartile (1)	−2.91	−6.02, 0.21	−2.93	−6.04, 0.19
3° quartile (2–3)	−0.01	−3.00, 2.97	0.15	−2.84, 3.14
4° quartile (>3)	−0.53	−3.63, 2.57	−0.45	−3.54, 2.63
Street markets				
No	Reference		Reference	
Yes	−0.11	−2.84, 2.63	−0.23	−2.96, 2.51

Values expressed in beta and 95% confidence interval (95% CI). Values in bold represent statistical significance. ^a^ Multilevel linear regression models with fixed effects. ^b^ Model adjusted for sex, age, skin color, education of the head of the household, and family income. ^c^ Model adjusted for sex, age, skin color, education of the head of the household, family income, body mass index, and physical activity level.

## Data Availability

The data presented in this study are available in the article.

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
