# Peer review of "Local Food Environment and Consumption of Ultra-Processed Foods: Cross-Sectional Data from the Nutritionists’ Health Study—NutriHS"

_ijerph, 2023, doi:10.3390/ijerph20186749_

Round 1
Reviewer 1 Report (New Reviewer)
This is an interesting study with well-constructed contents. I only have a few suggestions:
1. There are some typos in the manuscript, which should be carefully edited.
2. Why did you choose nutrition students and newly graduated nutritionists?
3. Can your findings provide some kind of guidance to non-nutrition-related groups?
Minor editing of English language is required.
Author Response
REBUTTAL LETTER
Date: August 16th, 2023
To: “International Journal of Environmental Research and Public Health”
Title: Local food environment and consumption of ultra-processed foods: data from the Nutritionist’ Health Study – NutriHS
Dear Editor,
Thank you for your consideration about our work. We appreciated the reviewers’ comments, which considerably improved our manuscript. Below please find our point-to-point responses to reviewers’ comments and modifications that have been made to the manuscript.
Review Report (round 1)
(1) There are some typos in the manuscript, which should be carefully edited.
Response: Thank you for your feedback. The manuscript has been thoroughly reviewed by a certified translator, and all previously requested changes have been implemented.
(2) Why did you choose nutrition students and newly graduated nutritionists?
Response: This approach was chosen to minimize follow-up losses in the NutriHS cohort study and to leverage a unique opportunity for gathering high-quality dietary information. This information has been included in the "2.1. Study Scenario and Sample" subsection of the methods, specifically in the first paragraph on page 3.
(3) Can your findings provide some kind of guidance to non-nutrition-related groups?
Response: The generalizability of the results of this study is limited for the non-nutrition-related groups, and even to non-nutrition-related groups, given the relatively small sample and specificities of the sample of nutrition students. This information has been incorporated into the discussion section, within the last paragraph on page 13.
Reviewer 2 Report (New Reviewer)
1. The STROBE guideline should be completed and attached as a supplement.
2. The cross-sectional design should be mentioned in the title.
Author Response
REBUTTAL LETTER
Date: August 16th, 2023
To: “International Journal of Environmental Research and Public Health”
Title: Local food environment and consumption of ultra-processed foods: data from the Nutritionist’ Health Study – NutriHS
Dear Editor,
Thank you for your consideration about our work. We appreciated the reviewers’ comments, which considerably improved our manuscript. Below please find our point-to-point responses to reviewers’ comments and modifications that have been made to the manuscript.
Review Report (round 1)
(1) The STROBE guideline should be completed and attached as a supplement.
Response: Thank you for your feedback. The STROBE guideline was completed and attached as a supplement.
(2) The cross-sectional design should be mentioned in the title.
Response: Thank you for your feedback. The study design was included in the title.

Reviewer 3 Report (New Reviewer)
This study provides a valuable insight into the relationship between the food environment and food intake, based on the NOVA classification, in Brazil. As a piece of public health nutrition research, it is quite intriguing and is sure to pique the interest of readers. While the paper does not have any major issues, addressing the following minor comments would be beneficial:
1. The sample size is not only small, but there also seems to be a considerable number of dropouts post-registration. Could this introduce any bias in the responses? Or should this merely be viewed as random missing data?
2. In Table 4, is the second tertile for supermarkets and hypermarkets also significant (bold) in Model 2?
3. Also, in Table 4, the second tertile is significant, yet the third tertile isn't and has a relatively small coefficient. What could be the possible reasons for this? It would be preferable if the discussion section elaborates on this point.
Author Response
REBUTTAL LETTER
Date: August 16th, 2023
To: “International Journal of Environmental Research and Public Health”
Title: Local food environment and consumption of ultra-processed foods: data from the Nutritionist’ Health Study – NutriHS
Dear Editor,
Thank you for your consideration about our work. We appreciated the reviewers’ comments, which considerably improved our manuscript. Below please find our point-to-point responses to reviewers’ comments and modifications that have been made to the manuscript.
Review Report (round 1)
(1) The sample size is not only small, but there also seems to be a considerable number of dropouts post-registration. Could this introduce any bias in the responses? Or should this merely be viewed as random missing data?
Response: Thank you for your feedback. There were no sociodemographic differences between the 734 participants who registered on the website and the 357 participants in the final sample, with data losses referring to randomly missing data. This information has been incorporated into the methods subsection titled "Study Scenario and Sample" within the final paragraph on page 3.
In addition, the final sample size is in accordance with the calculation of the minimum sample estimate at each center in the NutriHS study. This information was included in the second paragraph of the methods subsection "2.1. Study scenario and sample"
(2) In Table 4, is the second tertile for supermarkets and hypermarkets also significant (bold) in Model 2?
Response: Thank you for your feedback. Yes, the mentioned result was also significant. The correction was made in Table 4.
(3) Also, in Table 4, the second tertile is significant, yet the third tertile isn't and has a relatively small coefficient. What could be the possible reasons for this? It would be preferable if the discussion section elaborates on this point.
Response: Thank you for your feedback. These findings suggest that the relationship between food establishments and consumption of foods based on the level of processing consumption is non-linear, highlighting the need for further investigation and understanding of the underlying mechanisms. This information was included into the discussion section, in the last paragraph on page 14.
This manuscript is a resubmission of an earlier submission. The following is a list of the peer review reports and author responses from that submission.
Round 1
Reviewer 1 Report

I believe this paper was originally written in Portuguese. There was at least one part that was not translated into English and some translation were not quite right. It may be helpful to send the paper to a proofreader.
Author Response
REBUTTAL LETTER
Date: May 26th, 2023
To: “International Journal of Environmental Research and Public Health”
Title: Local food environment and consumption of ultra-processed foods: data from the Nutritionist’ Health Study – NutriHS
Dear Editor,
Thank you for your consideration about our work. We are re-submitting the manuscript with the requested modifications.
We appreciated the reviewers’ comments, which considerably improved our manuscript.
Below please find our point-to-point responses to reviewers’ comments and modifications
that have been made to the manuscript.
Review report (Round 1)
Major issues
- 1. The authors use quartiles or tertiles for food environment categories. This categorization seems questionable, given some categories have very low numbers (e.g., median of 1 with p25-75 range of 0- 2). Some of the “stronger” evidence they present have very wide 95% confidence intervals, which makes me wonder if the n in those quartiles were very small. It would be better to conduct the analyses using just the number of each type of food outlets within each buffer – if the data are skewed (e.g., many 0’s), you need to consider other types of models to fit.
Response: Thank you for your observations regarding the categorization of food environments in our study. However, we would like to explain the rationale behind our choice to use quartiles or tertiles as a categorization method.
The use of quartiles or tertiles for categorizing food environments was a decision made based on theoretical and practical considerations. This approach allows us to examine the distribution of food outlets in relation to the study population, identifying patterns and potential associations.
We acknowledge that some categories may have low numbers, as indicated by the median of 1 and the range of 0-2 in the 25th-75th percentile. However, it is important to note that the presence of categories with low numbers is a characteristic of the actual data available for our study. These categories may represent specific situations or geographical areas with limited availability of food outlets.
We understand your suggestion to consider the number of each type of food outlet within each buffer zone as an alternative approach. However, we would like to clarify that the quartiles or tertiles used for categorization in our study were not based on small sample sizes. The confidence intervals are also influenced by the variability of the data points within those categories.
We understand your suggestion to consider the absolute number of each type of food outlet within each buffer zone. However, in doing so, we would be losing valuable information about the relative distribution of food outlets within the analyzed population.
We appreciate your observations and comments and respect your concerns. However, we believe that the used categorization is appropriate for the objectives and scope of our study.
- 2. The abstract says “The presence of mini-markets in the buffer was associated with lower consumption of ultra-processed foods, when adjusted for socioeconomic and lifestyle conditions (β = - 3.29; 95%CI = - 6.39 to - 0.19). The presence of bakeries and cafeterias was related to lower consumption of ultra-processed foods among participants, when adjusted for socioeconomic conditions (β = -3.10; 95%CI = -6.18 to -0.02).” However, this is not exactly what’s presented in the results section. The authors conducted the analyses using quartiles/tertiles and these coefficients were from one of the quartiles – in fact, they saw evidence of association in one of the middle quartiles but not in the highest quartile.
Response: Thank you for the considerations. The information that the findings correspond to the third quartile has been added to the abstract (line 17).
- 3. Overall, what I see is that there was no clear evidence of association between any types of food outlets and UPF consumption—which is a fine conclusion and worth publishing. The few associations we see (e.g., 3rd quartile of bakeries/cafeterias and UPF) may just be statistical artifacts.
Response: Thank you for the considerations. This information was included in the first paragraph of the conclusion and in the abstract.
- 4. The authors mention the strengths of conducting this study with nutrition students. The authors need to also discuss the weaknesses, especially the extent to which the findings could be generalized.
Response: The limitation in the generalization of the results of this study, as it was carried out with Nutrition students, was added to the limitations of the study (from line 403).
Minor issues
- 1. I believe this paper was originally written in Portuguese. There was at least one part that was not translated into English and some translation were not quite right. It may be helpful to send the paper to a proofreader.
Response: We agree with your suggestion. The manuscript underwent proper English language review by a professional.
- 2. In the abstract, please include more information about the study population’s characteristics (e.g., sample size).
Response: The sample size information was included in the abstract.
- 3. In the Methods section, you mention the power calculation but presumably, that was done for other outcomes (the main study’s outcomes) and that information is not relevant here.
Response: Thank you for the considerations. This information was removed from the methods.
- 4. It’d be good to justify why you eliminated those with less than 500kcal and greater than 5000kcal.
Response: Thank you for the considerations. The justification was included in the methods and the reference used was added to the body of the text and to the list of references.
- 5. The authors need to add a little more explanation about context specific information for the readers. For example, you should explain what “Municipal Human Development Index (MHDI) of 0,754” means – explain what MHDI indicates and how the readers should interpret the number 0.754. Another example is minimum wages – what does 6-10 minimum wages mean?
Response: The explanation of the meaning of the HDI was included in the third paragraph of the methods, as well as the reference to the minimum wage in Brazil.
- 6. It’d be better to use overweight/obese and not overweight/obese unless there was no participant whose BMI was in the obese range (>30kg/m2) – if there was no participant with obesity, please insert that sentence when you talk about anthropometry.
Response: The adjustments were made to the methods and Table 1.
Reviewer 2 Report
Summary
This paper examines the relationship between ultra-processed food consumption and the local food environment in a small sample of Nutrition students in Brazil. The authors found that higher levels of mini markets in a student’s local environment was associated with lower consumption of ultra-processed food. The study uses a novel combination of data sources to answer an interesting and valid research question. However, I feel there are several limitations which need to be addressed before the paper is published
Major points
(1) The study uses a small sample of nutrition students in one city in Brazil. This is a very specific group, as such the conclusions are very limited and cannot be generalised more widely. I think it is important to highlight this limitation more frequently when reporting and discussing the findings. For example, the authors do not remind the reader that the population is a sample of Nutrition students in the first paragraph of the discussion. The reader may misunderstand and think this could be generalised to the wider population. Additionally, the limited generalisability is not discussed in the limitations.
(2) The authors report an inverse association between the 3rd quartile of density of bakeries and coffee shops and the caloric contribution of ultra-processed foods. However, there is a lot of uncertainty around this finding that is not discussed. Firstly. the tables show that this does not remain significant after adjusting for BMI and physical activity. Secondly, there is no association 4th quartile (highest) compared to the 1st quartile, which you would expect if there was a true association between the density of bakeries in a local area and ultra-processed food consumption. The authors do not discuss these two aspects and therefore it seems that they are misrepresenting a finding for which there should be a high degree of caution.
(3) In the discussion, the authors suggest that the finding between the density of bakeries and consumption ultra-processed food can be explained as the participant’s nutritional knowledge will affect their food choice. However, aside from the reasons in comment 2, this explanation does not seem appropriate. If I understand correctly, the finding suggests that nutrition students with a higher number of bakeries in their local environment (3rd quartile) eat less ultra-processed food in their diet compared to nutrition students with the lowest amount of bakeries in their local environment (1st quartile). As all the participants have had the same nutritional education, there’s no evidence to suggest that those who live near more bakeries would use their knowledge to make different food choices than those who live near less.
(4) The categorisation of the food outlets may introduce bias into the study. Firstly, I don’t think that the grouping of retail stores into the groups “retail of unprocessed and minimally processed foods” and “retail of processed and ultra-processed foods” is appropriate as it uses the outcome variable to define categories of the exposure variable. Additionally, it is both unclear and unlikely that these retail stores would sell only minimally processed or processed/ultra-processed foods, resulting in misclassification bias. Secondly, there is not a clear distinction between the categories. For example, “bakeries and coffee shops” and “snack bars”, appear to be very similar. Ice-cream is also mentioned in the “ultra-processed retail” and the “snack bar” category. It could both simplify the findings and reduce misclassification bias to use the following categories which are grouped more by purpose and size of the business: 1) supermarkets and hypermarkets, 2) small markets (minimarkets, grocery stores, warehouses, including small business selling food such as meat, fish vegetables and bread etc.), 3) street markets, 4) restaurants, 5) bars and bakeries (snack bars, coffee shops, bakeries, tea houses, juice shops, pastry and ice cream shops).
(5) The discussion could be improved to have a more thorough examination of the findings and relate these to the wider literature on environmental influences on the diet. For example, there is no mention of other studies which have looked at the density of different types of food outlets with diet and whether this study supports those conclusions. Overall, does this study support the hypothesis that local environment impacts diet? How confident are the authors in this finding? Is it similar to other research on the topic?
(6) The conclusion section has many statements which are not supported by the evidence from the paper. The authors suggest a causal relationship between the food environment and food consumption which cannot be inferred from this cross-sectional study. Also, the authors mention food insecurity, which has not been mentioned previously and was not directly studied.
Minor points
(7) Lines 34-36: The authors do not explain or define ultra-processed food. It is important to give a brief description of how ultra-processed food differs from processed food in the introduction.
(8) Lines 61:64: This sentence needs to be rephrased as the meaning is not clear.
(9) Line 112: This is quite a high drop out rate, the impact of small sample size needs to be discussed throughout the paper, especially in the limitation section
(10) Lines 122-123: I am aware that it this grouping of ethnicities is common in Brazil and that “Amarello” is used in official category used, however it does not translate well into English. I think it would be better to use “East Asian” or something along those lines.
(11) Lines 128-129: Self-reported data height and weight data is likely to be biased. This needs to be discussed in the limitation section
(12) Lines 156-157: It would be helpful if the authors provided the categorisation of the food by the NOVA groups in the supplementary files so the reader could assess this categorisation.
(13) Line 210: The phrase “the amount of equipment” is not the best translation. I think the authors might mean “the number of stores/establishments”.
(14) Lines 216-19: The authors categorise the number of stores/establishments into quartiles and tertiles. It would be helpful if the number of stores in each quartile is described so the reader can better understand the results.
(15) Table 1: There are some phrases in this table which need to be translated to English. Also, the “Regional” variable which ranges from 1-6 needs to be better explained.
(16) The units for the NOVA categories are not given, it is not clear that this related to percentage of total calories. This should be applied throughout the results.
(17) Figure 3: too small, the text is unreadable.
(18) Table 2: The formatting of the table needs to be changed so that the 1st -4th quartiles are not also in bold so that it is clear they are sub-levels. Change “IC” to “CI” for the correct abbreviation of “confidence intervals”. Also, between the lower and higher confidence limits ‘a’ needs to be replaced with “-“or “,”.
(19) Lines 259-264: I don’t think that the results for culinary ingredients add much to the paper and could be removed.
(20) Lines 319-321: I think this misrepresents the findings and should be revised, see comment 2.
The use of English needs to be carefully reviewed. The tables were not fully translated. Additionally many phrases were not translated well so were confusing (see comments above)
Author Response
REBUTTAL LETTER
Date: May 26th, 2023
To: “International Journal of Environmental Research and Public Health”
Title: Local food environment and consumption of ultra-processed foods: data from the Nutritionist’ Health Study – NutriHS
Dear Editor,
Thank you for your consideration about our work. We are re-submitting the manuscript with the requested modifications.
We appreciated the reviewers’ comments, which considerably improved our manuscript.
Below please find our point-to-point responses to reviewers’ comments and modifications
that have been made to the manuscript.
Review report (Round 1)
Major points
- The study uses a small sample of nutrition students in one city in Brazil. This is a very specific group, as such the conclusions are very limited and cannot be generalised more widely. I think it is important to highlight this limitation more frequently when reporting and discussing the findings. For example, the authors do not remind the reader that the population is a sample of Nutrition students in the first paragraph of the discussion. The reader may misunderstand and think this could be generalised to the wider population. Additionally, the limited generalisability is not discussed in the limitations.
Response: Thank you for your suggestions. An excerpt about the limitation of the study sample was included among the limitations of the study (from line 396).
- The authors report an inverse association between the 3rd quartile of density of bakeries and coffee shops and the caloric contribution of ultra-processed foods. However, there is a lot of uncertainty around this finding that is not discussed. Firstly, the tables show that this does not remain significant after adjusting for BMI and physical activity. Secondly, there is no association 4th quartile (highest) compared to the 1st quartile, which you would expect if there was a true association between the density of bakeries in a local area and ultra-processed food consumption. The authors do not discuss these two aspects and therefore it seems that they are misrepresenting a finding for which there should be a high degree of caution.
Response: A comment regarding this interpretation of the results was added as the sixth paragraph of the discussion.
- In the discussion, the authors suggest that the finding between the density of bakeries and consumption ultra-processed food can be explained as the participant’s nutritional knowledge will affect their food choice. However, aside from the reasons in comment 2, this explanation does not seem appropriate. If I understand correctly, the finding suggests that nutrition students with a higher number of bakeries in their local environment (3rd quartile) eat less ultra-processed food in their diet compared to nutrition students with the lowest amount of bakeries in their local environment (1st quartile). As all the participants have had the same nutritional education, there’s no evidence to suggest that those who live near more bakeries would use their knowledge to make different food choices than those who live near less.
Response: We welcome comments regarding our manuscript. As stated in the response to comment 2, we reiterate our recognition that the interpretation of our results does not follow a response linearity. We added a comment about it in the sixth paragraph of the discussion.
- The categorisation of the food outlets may introduce bias into the study. Firstly, I don’t think that the grouping of retail stores into the groups “retail of unprocessed and minimally processed foods” and “retail of processed and ultra-processed foods” is appropriate as it uses the outcome variable to define categories of the exposure variable. Additionally, it is both unclear and unlikely that these retail stores would sell only minimally processed or processed/ultra-processed foods, resulting in misclassification bias. Secondly, there is not a clear distinction between the categories. For example, “bakeries and coffee shops” and “snack bars”, appear to be very similar. Ice-cream is also mentioned in the “ultra-processed retail” and the “snack bar” category. It could both simplify the findings and reduce misclassification bias to use the following categories which are grouped more by purpose and size of the business: 1) supermarkets and hypermarkets, 2) small markets (minimarkets, grocery stores, warehouses, including small business selling food such as meat, fish vegetables and bread etc.), 3) street markets, 4) restaurants, 5) bars and bakeries (snack bars, coffee shops, bakeries, tea houses, juice shops, pastry and ice cream shops).
Response: The last two establishments (“retail of unprocessed and minimally processed foods” and “retail of processed and ultra-processed foods”) were grouped according to the predominance of acquisition by the Brazilian population and food processing characteristics, according to the classification proposed by the Interministerial Chamber for Food and Nutrition Security (CAISAN). Regarding the classification of ice cream shops, we have wholesale, which sells a variety of food products, such as ice cream, being classified as (retail and wholesale of processed and ultra-processed foods). And we have ice cream shops, which were classified as snack bars, for categorizing the sale ready-to-eat food products in retail, being, therefore, a different category from the one presented for retail. To facilitate differentiation, we have included the word "stores" in the ice cream naming in the snack bars category.
- The discussion could be improved to have a more thorough examination of the findings and relate these to the wider literature on environmental influences on the diet. For example, there is no mention of other studies which have looked at the density of different types of food outlets with diet and whether this study supports those conclusions. Overall, does this study support the hypothesis that local environment impacts diet? How confident are the authors in this finding? Is it similar to other research on the topic?
Response: A paragraph was added in the discussion (from line 384) with results from another study that evaluated the relationship between the food environment and the consumption of ultra-processed foods. In our study, we found significant data to support the internal validity of the results and support the conclusion that the presence of some establishments influenced the caloric contribution of some food groups analyzed.
- The conclusion section has many statements which are not supported by the evidence from the paper. The authors suggest a causal relationship between the food environment and food consumption which cannot be inferred from this cross-sectional study. Also, the authors mention food insecurity, which has not been mentioned previously and was not directly studied.
Response: The excerpt from the conclusion related to food insecurity has been removed. Our conclusion was reformulated to suit the results found and the study design.
Minor points
- Lines 34-36: The authors do not explain or define ultra-processed food. It is important to give a brief description of how ultra-processed food differs from processed food in the introduction.
Response: Thank you for your consideration. The explanation of the ultra-processed food classification was included in the first paragraph of the introduction.
- Lines 61:64: This sentence needs to be rephrased as the meaning is not clear.
Response: Thank you for your suggestion. The sentence has been reformulated to improve its understanding.
- Line 112: This is quite a high drop out rate, the impact of small sample size needs to be discussed throughout the paper, especially in the limitation section.
Response: Thank you for your consideration. The sample was inserted as a limitation of the study at the end of the discussion (from line 403).
- Lines 122-123: I am aware that it this grouping of ethnicities is common in Brazil and that “Amarello” is used in official category used, however it does not translate well into English. I think it would be better to use “East Asian” or something along those lines.
Response: Thank you for your consideration. We adjusted the ethnic classification and replaced "yellow" with "east Asian" in topic 2.2.1. of the methods, related to Sociodemographic characteristics.
- Lines 128-129: Self-reported data height and weight data is likely to be biased. This needs to be discussed in the limitation section.
Response: The use of self-reported weight and height information was included as a study limitation at the end of the discussion (from line 403).
- Lines 156-157: It would be helpful if the authors provided the categorization of the food by the NOVA groups in the supplementary files so the reader could assess this categorization.
Response: Thank you for your consideration. On the indicated lines, we include an excerpt that mentions the reference used in the study for categorization of foods according to the NOVA classification, where readers can find more information.
- Line 210: The phrase “the amount of equipment” is not the best translation. I think the authors might mean “the number of stores/establishments”.
Response: Thank you for your consideration. We have made the requested changes in the indicated lines.
- Lines 216-19: The authors categorise the number of stores/establishments into quartiles and tertiles. It would be helpful if the number of stores in each quartile is described so the reader can better understand the results.
Response: Thank you for your suggestions. The number of establishments/food stores was added to each quartile or tertile cited in the tables.
- Table 1: There are some phrases in this table which need to be translated to English. Also, the “Regional” variable which ranges from 1-6 needs to be better explained.
Response: More information about "Regional" has been included in the third paragraph of the methods.
- The units for the NOVA categories are not given, it is not clear that this related to percentage of total calories. This should be applied throughout the results.
Response: Thank you for your suggestion. We added information regarding the analysis of the NOVA classification in percentage of total energy in table 1.
- Figure 3: too small, the text is unreadable.
Response: Thank you for your suggestion. We have improved the image resolution quality and increased the font size of subtitles.
- Table 2: The formatting of the table needs to be changed so that the 1st-4thquartiles are not also in bold so that it is clear they are sub-levels. Change “IC” to “CI” for the correct abbreviation of “confidence intervals”. Also, between the lower and higher confidence limits ‘a’ needs to be replaced with “-“or “,”.
Response: Thank you for your considerations. The adjustments in the formatting of the tables were made.
- Lines 259-264: I don’t think that the results for culinary ingredients add much to the paper and could be removed.
Response: Results regarding culinary ingredients have been removed.
- Lines 319-321: I think this misrepresents the findings and should be revised, see comment 2.
Response: Thank you for your considerations. A commentary on this subject was included in the discussion (from line 375).
Round 2
Reviewer 2 Report
Dear Authors,
I am grateful for the time you have spent on responding to my comments and revising to the paper. I agree that the paper has been improved in many ways, including some translation of the paper, some edits to conclusions and repeating the study's population more frequently. However, in many ways the edits to the paper have not been thorough enough and I do not feel the paper is ready for publication.
(1) In my first review, I commented that the inconsistency of the association found suggested there was not strong evidence for an association. In response, the authors have edited the manuscript in places to describe this more accurately. However the issue is that the text has been inserted into the manuscript rather than more broadly looking at revising the tone and direction of the whole manuscript. As such there are many seemingly contradictory statements. For example, the first paragraph of the of the discussion does not indicate a there was not a strong evidence for an association. Also in the fourth paragraph of the discussion a similar statement is made. Additionally in the conclusion section, it says both that the study found an association between some store types and that there was no clear evidence? The whole paper needs to be revised to ensure a consistent message is given throughout. Lastly, I do not think it is appropriate that the authors cut-and-pasted sections of my review into the manuscript.
(2) While I appreciate that the grouping of stores is a difficult task, I still remain unconvinced by the authors approach. For example, I do not see a clear distinction between group 5 (snack bars) and group 6 (bakeries and coffee shops). Minimal changes to the manuscript have been made in results of my comments. Furthermore, the authors quote a paper with different, but more understandable groupings of establishments. Is there a reason that the authors did not use the same approach as the paper cited? (Costa et al. 2013).
(3) In my last review I mentioned that it is not logical to use the participant's nutrition knowledge as an explanation for the findings in this study as all the participants have the same level of nutrition knowledge. The authors did not acknowledge this point in their response nor edited the manuscript accordingly. Therefore, I still have an issue with this section of the manuscript.
The manuscript still needs careful revision to the English, which I do not feel has been done appropriately yet. In my last review I highlighted incorrect usage of the word "equipment". The authors edited that one instance of the word rather than reviewing the whole document. There are other instances where the English needs to be improved, however I do not feel it is my role to highlight them all.